# OBPOSE: LEVERAGING POSE FOR OBJECT-CENTRIC SCENE INFERENCE AND GENERATION IN 3D

## ABSTRACT

We present OBPOSE, an unsupervised object-centric inference and generation model which learns 3D-structured latent representations from RGB-D scenes. Inspired by prior art in 2D representation learning, OBPOSE considers a factorised latent space, separately encoding object location (*where*) and appearance (*what*). OBPOSE further leverages an object's *pose* (i.e. location and orientation), defined via a minimum volume principle, as a novel inductive bias for learning the *where* component. To achieve this, we propose an efficient, voxelised approximation approach to recover the object shape directly from a neural radiance field (NeRF). As a consequence, OBPOSE models each scene as a composition of NeRFs, richly representing individual objects. To evaluate the quality of the learned representations, OBPOSE is evaluated quantitatively on the YCB, MultiShapeNet, and CLEVR datatasets for unsupervised scene segmentation, outperforming the current state-of-the-art in 3D scene inference (ObSuRF) by a significant margin. Generative results provide qualitative demonstration that the same OBPOSE model can both generate novel scenes and flexibly edit the objects in them. These capacities again reflect the quality of the learned latents and the benefits of disentangling the *where* and *what* components of a scene. Key design choices made in the OBPOSE encoder are validated with ablations.

## 1 INTRODUCTION

In recent years, object-centric representations have emerged as a paradigm shift in machine perception. Intuitively, inference or prediction tasks in down-stream applications are significantly simplified by reducing the dimensionality of the hypothesis space from raw perceptual inputs, such as pixels or point-clouds, to something more akin to a traditional state-space representation. While reasoning over objects rather than pixels has long been the aspiration of machine vision research, it is the ability to learn such a representation in an unsupervised, generative way that unlocks the use of large-scale, unlabelled data for this purpose. As consequence, research into object-centric generative models (OCGMs) is rapidly gathering pace.

Central to the success of an OCGM are the inductive biases used to encourage the decomposition of a scene into its constituent components. With the field still largely in its infancy, much of the work to date has confined itself to 2D scene observations to achieve both scene inference (e.g. Eslami et al., 2016; Burgess et al., 2019; Greff et al., 2016; 2017; Locatello et al., 2020; Engelcke et al., 2019; 2021) and, in some cases, generation (e.g. Engelcke et al., 2019; 2021). In contrast, unsupervised methods for object-centric scene decomposition operating directly on 3D inputs remain comparatively unexplored (Elich et al., 2022; Stelzner et al., 2021) – despite the benefits due to the added information contained in the input. As a case in point, Stelzner et al. (2021) recently established that access to 3D information significantly speeds up learning. Another benefit is that, for the parts of an object visible to a 3D sensor, object *shape* is readily accessible and does not have to be inferred, either from a single view (e.g. Liu et al., 2019; Kato et al., 2018) or from multiple views (e.g. Yu et al., 2021; Xie et al., 2019). We conjecture that object shape can serve as a highly informative inductive bias for object-centric learning. As we elaborate below, we reason that the asymmetry of a shape can be used to discover an object's pose, and pose can help to identify and locate an object in space.

Here we present OBPOSE, an unsupervised OCGM that takes RGB-D images (or video) as input and learns to segment the underlying scene into its constituent 3D objects, as well as into an explicit background representation. As we will show, OBPOSE can also be used for scene generation and editing. Inspired by prior art in 2D settings (Eslami et al., 2016; Crawford & Pineau, 2019; Lin et al., 2020; Kosiorek et al., 2018; Jiang et al., 2019; Wu et al., 2021), OBPOSE factorises its

latent embedding into a component capturing an object's location and appearance (*where* and *what* components, respectively). This factorisation provides a strong inductive bias, helping the model to disentangle its input into meaningful concepts for downstream use.

A key contribution of OBPOSE is the introduction of *pose* (i.e. location and orientation) as a novel inductive bias.

OBPOSE is not a pose-estimation model. Rather, OBPOSE infers pose information from an object's shape to reduce apparent variance and to simplify the learning of the model's *what* component in 3D (see fig. 11 in the appendix) – ultimately for use in downstream tasks like segmentation and scene editing. In an OGGM, the intuition is that the variance we observe in an object's pose should be made invariant in the latent space. OBPOSE infers pose information without supervision, using a minimum volume principle defined using the tightest bounding box that constrains the object. Effectively, the tightest bounding box will reveal asymmetries in an object's shape, if there are any, which can be used to constrain the object's orientation. We further propose a voxelised approximation approach that recovers an object's shape in a computationally tractable way from a neural radiance field (NeRF) (Mildenhall et al., 2020). Although the recovery of an object's shape from a NeRF can be prohibitively expensive, our approach allows this to be integrated efficiently into the OBPOSE training loop.

In a series of experiments, OBPOSE outperforms the current state-of-the-art in 3D scene inference, ObSuRF (Stelzner et al., 2021), by significant margins. Evaluations are performed on the CLEVR dataset (Johnson et al., 2017), the MultiShapeNet dataset (Stelzner et al., 2021; Chang et al., 2015), and the YCB dataset for unsupervised scene segmentation (Calli et al., 2015), in the latter case using both RGB-D moving-objects (video) and multi-view static scenes. An ablation study on the OBPOSE encoder serves to validate the design decisions that distinguish its use of attention from alternative attention mechanisms represented by Slot Attention (Locatello et al., 2020) and GENESIS-v2 (Engelcke et al., 2021). In summary, the key contributions of this paper are: (1) a new state-of-the-art unsupervised scene segmentation model for 3D, OBPOSE, together with insights into its design decisions; (2) a novel inductive bias for 3D OCGMs, *pose*, together with its motivation; and (3) a general method for fast shape evaluation from NeRFs.

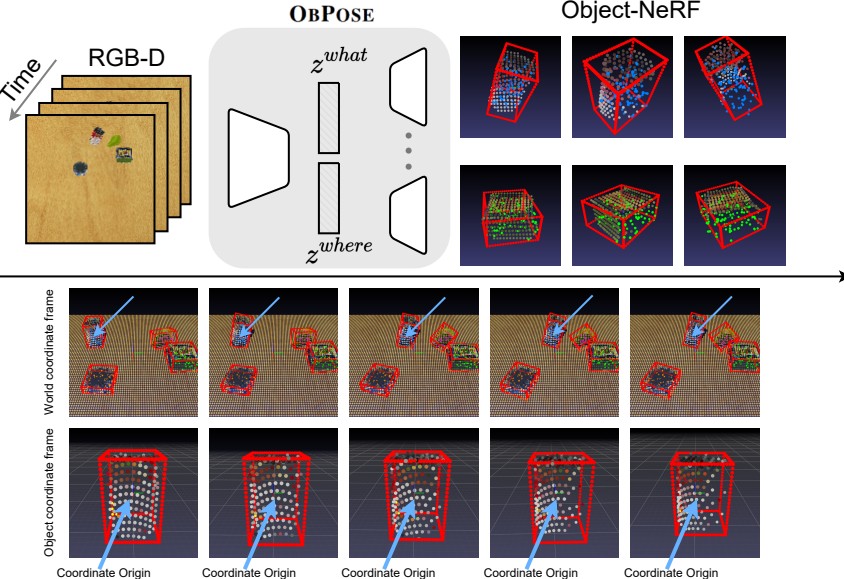

Figure 1: Overview. Front-view observations can be fed into OBPOSE (e.g. video of objects on a table, top left). OBPOSE then reconstructs the sparse voxelised point cloud and normalised *pose* of each object (two objects shown, top right). Each point within a slot is coloured to represent a specific object. Note that while an object may move in the world frame (middle row), it appears stationary when viewed from the perspective of its normalised frame (bottom row).

## 2 METHODS

OBPOSE takes RGB-D videos (or images) as input and learns to segment scenes into a set of foreground objects, with a single background component. The location and orientation of objects can be estimated from the respective shapes which we reconstruct using NeRFs (Mildenhall et al., 2020). In contrast to previous works (Stelzner et al., 2021) where object NeRFs have to model object appearance and location jointly, each object NeRF in OBPOSE only models the 3D geometry and the texture of the objects by conditioning on the predicted object locations and orientations.To perform location and orientation conditioned inference, OBPOSE clusters pointwise embeddings that are encoded from a standard (KPConv-based) backbone (Thomas et al., 2019) into a soft attention mask for each object using an instance colouring stick-breaking process (IC-SBP). Then a *where*-inference step predicts the object locations and orientations given the object-wise attention masks from the IC-SBP. Finally, a *what*-inference step encodes the appearance and shape information (conditioned on the object locations and orientations). This information is then decoded into NeRFs and all of the NeRFs are composed with the background component to reconstruct the original scene. A schematic overview of the OBPOSE architecture is provided in Figure 2.

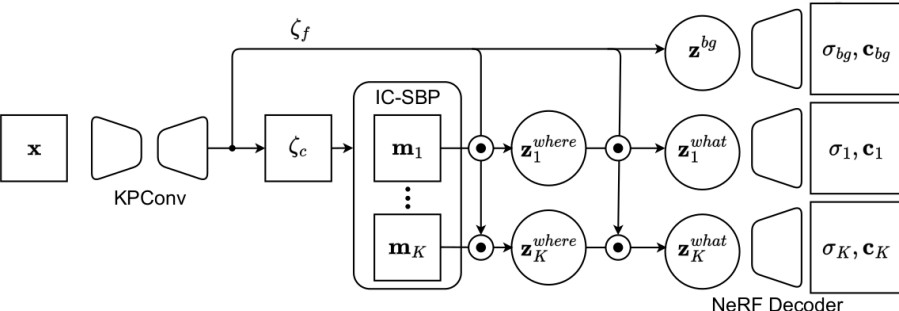

Figure 2: Model architecture. Given an input RGB-D image $\mathbf{x}$, the KPConv backbone extracts point embeddings $\zeta_c$ and a background component $\mathbf{z}^{bg}$. The point embeddings are clustered by the IC-SBP into soft attention masks, $\mathbf{m}_1 \ldots \mathbf{m}_K$, for each object slot. Given these object masks, a *where* module infers the location of each object encoded by $\mathbf{z}^{where}$. Conditioned on object poses, which are recovered via the minimum volume principle, the *what* latents, $\mathbf{z}^{what}$, are encoded and then decoded into object NeRFs. Object NeRFs are finally composed with the background component to reconstruct the observed scene.

### 2.1 ENCODER

The input observation $\mathbf{x}_t$ is a RGB-D image of height $H$ and width $W$ for each time-step $t \in \{1 \ldots T\}$. RGB-D depth images are converted into the point clouds $\mathbf{p}_t$ using the known camera parameters (extrinsic and intrinsic). The point clouds $\mathbf{p}_t$ and the RGB channels of $\mathbf{x}_t$ are then concatenated and encoded into a point embedding $\zeta_t^{H_1 \times W_1 \times D}$. This is achieved using a U-Net-like backbone module (Ronneberger et al., 2015) consisting of several KPConv layers (Thomas et al., 2019). A KPConv layer is an extension of a 2D convolutional neural network (CNN) layer for point clouds, which preserves the translation invariance properties of CNNs. For computational efficiency, output embeddings are upsampled to the resolution of the first downsampled embedding using trilinear interpolation. An aggregation layer is required to summarise the global information for encoding the background information into the latent representation $\mathbf{z}^{bg}$. For classification tasks, average pooling was adopted for aggregation (Thomas et al., 2019). However, the pooling is not suitable for capturing the location information of the input features. Instead, we use a KPConv-aggregate layer to aggregate the output features, initialising a convolutional kernel at the coordinate centre. In this way, global spatial information is preserved and encoded in the relative distances between the output features and the kernel points. Following prior work (Engelcke et al., 2021), we additionally build two heads (point-wise MLPs) upon the output embedding $\zeta_t$, with one predicting the colour embedding $\zeta_{c,t}^{H_1 \times W_1 \times D_c}$ for the IC-SBP, and the other the feature embedding $\zeta_{f,t}^{H_1 \times W_1 \times D_f}$ for other encoding tasks.

### 2.1.1 Instance Colouring Stick-Breaking Process for Video

An instance colouring stick-breaking process (IC-SBP) is a clustering algorithm that takes point embeddings $\zeta_{c,t}^{H_1 \times W_1 \times D_c}$ as inputs and outputs $K$ predicted soft attention masks $\mathbf{m}_k \in [0,1]^{H_1 \times W_1}$. The stick-breaking process (Burgess et al., 2019) guarantees that the masks are normalised:

$$\mathbf{m}_1 = \alpha_1, \quad \mathbf{m}_k = \mathbf{s}_{k-1} \odot \alpha_k, \quad \mathbf{m}_K = \mathbf{s}_K, \tag{1}$$

where the *scope* $\mathbf{s}_k \in [0,1]^{H_1 \times W_1}$ tracks which pixels have not yet been explained. $\mathbf{s}_k$ is initialised and updated as follows:

$$\mathbf{s}_0 = \mathbb{1}^{H_1 \times W_1}, \quad \mathbf{s}_k = \mathbf{s}_{k-1} \odot (1 - \alpha_k) \tag{2}$$

The alpha mask $\alpha_k$ is computed as the distance between a cluster seed $\zeta_{i,j}$ and all individual point embeddings according to a kernel $\psi$. Readers are referred to (Engelcke et al., 2021) for details on kernel selection; our implementation uses Gaussian kernels. The IC-SBP prevents the model from learning a fixed order of decomposition. It achieves this by selecting the cluster seed at the spatial location $(i,j)$ according to the argmax of a score mask $\mathbf{s}_k^c = \mathbf{s}_k \odot \mathbf{u}$ with $\mathbf{u}$ being a score map sampled from a uniform distribution $U(0,1)^{H_1 \times W_1}$. The score map $\mathbf{u}$ ensures the stochastic ordering of the mask $\mathbf{m}_k$.

Although the IC-SBP is by default agnostic with respect to the slots, there are nonetheless special components in scene segmentation such as the background. We model slots like these that contain special components by passing the pixel embeddings $\zeta_{c,t}^{H_1 \times W_1 \times D_c}$ through a multilayer perceptron (MLP) $\rho$ to compute a pre-scope $\mathbf{s}^p$:

$$\mathbf{s}^p = \operatorname*{softmax}_{K'}(\rho(\zeta) \in \mathbb{R}^{H_1 \times W_1 \times K'}) \tag{3}$$

Here $K'-1$ is the number of special slots. By convention, we take the last channel of $\mathbf{s}^p$ to be the *scope* $\mathbf{s}_0$ in the IC-SBP. In the present study, we set $K'=2$ for only modelling the background. A termination mechanism is implemented by sampling from the remaining scope $\mathbf{s}_k$ before computing the alpha mask $\alpha_k$ at each step $k$. Each point in the scope $\mathbf{s}_k$ is treated as an independent Bernoulli distribution with the probabilities being the scope value and perform point-wise sampling. The current slot is set to the idle state when all sampled values are zero, which indicates that the whole observation has been explained by previous slots.

For the video input, we extend the original IC-SBP to include an additional propagation step. The cluster seed $\zeta_{i,j}$ can be used as an ID for each slot throughout the video. The cluster seed sampled at $t=1$ is stored and used to compute the alpha masks for frames of $t \geq 2$. The idle slots detected at $t=1$ will remain in idle states during propagation. To ensure the normalisation of the masks $\mathbf{m}_k$, the SBP operation takes the remaining scope $\mathbf{s}_K$ as the last component mask $\mathbf{m}_K$. This does not have an associated seed for propagation. We run one more step of the SBP for the last component and flag the remaining scope $\mathbf{s}_K$ as unused. Concretely, we add an additional penalty loss to encourage the final remaining scope $\mathbf{s}_K$ to be zero everywhere, thereby motivating the model to explain the whole observation using only the previous slots.

### 2.1.2 Object Location and Orientation Conditioned Encoding

To facilitate a disentangled encoding of *what* and *where*, we firstly introduce shape-based object location and orientation estimation. We note that a point cloud $\mathbf{P}_{t,k} = \{\mathbf{p}_j | j \in \{1, ..., N_k\}\}$ of object $k$ at time step $t$ can be viewed as a discrete sampling from the object surface and thus preserves the shape information of the objects. Intuitively, the location of the object $\mathbf{T}_{t,k}$ can be initialised at the centre of mass of $\mathbf{P}_{t,k}$:

$$\mathbf{T}_{t,k} = \frac{1}{N_k} \sum_{j \in \{1, ..., N_k\}} \mathbf{p}_j \tag{4}$$

with $N_k$ being the number of points of object $k$. For the orientation of the object, which can be represented by a rotation matrix $\mathbf{R}_{t,k} \in SO(3)$, we propose the following *minimum volume principle* to define a unique shape-based object orientation: given a set of points, we set the orientation of the object represented by those points to the orientation of the tightest bounding box that contains those points. Concretely, we find this bounding box by first transforming the points under several selection rotation matrices $\mathbf{R}^s \in SO(3)$ and then computing the volume of the axis-aligned bounding boxes (AABBs) that contain these points. Selection rotation matrices are generated as equivolumetric grids on the $SO(3)$ manifold (Murphy et al., 2021; Yershova et al., 2010). This method is based on the

HEALPix method of generating equal area grids on the 2-sphere (Gorski et al., 2005). Due to the symmetry property of the bounding boxes, the smallest bounding box can have multiple solutions, e.g. swapping the x-axis and the y-axis of the coordinates of the bounding box will result in another bounding box that has the same volume. Therefore, we select all the bounding boxes whose volumes are in the range $[V_{\min}, (1 + \beta) V_{\min}]$ to account for this issue, where $V_{\min}$ is the minimum volume of all the AABBs and $\beta$ is a small tolerance factor. We pick the AABB whose orientation is closest to the world coordinates and whose orientation is represented by the identity matrix $\mathbf{I} \in \mathbb{R}^{3 \times 3}$ for the first time step, and otherwise to the orientation in the previous time step. The distance $d_{\mathrm{SO}(3)}$ is measured by the geodesic distance on the SO(3) manifold:

$$d_{\mathrm{SO}(3)} = \arccos\left(\frac{\mathrm{tr}(\mathbf{R}_{t,k}\mathbf{R}_{t-1,k}^{-1}) - 1}{2}\right) \tag{5}$$

The points $\mathbf{p}_j \in \mathbf{P}_{t,k}$ are then transformed to their object coordinates given a pair of pose parameters $\{\mathbf{T}, \mathbf{R}\}$ as follows:

$$k\left(\mathbf{p}_j, \mathbf{R}, \mathbf{T}\right) = \frac{2}{s}(\mathbf{R})^{-1}\left(\mathbf{p}_j - \mathbf{T}\right) \tag{6}$$

The bounding box size $s \in \mathbb{R}$ is initialised to a sufficiently large number for all objects. All points that are out of the bounding box are discarded so that the spatial coordinate of each point is in the range of $[-1, 1]$.

For each object slot at time step $t$, we find its associated point cloud $\mathbf{P}_{t,k}$ by computing the argmax over the attention masks $\mathbf{m}_{t,k}$ predicted from the IC-SBP. The point features $\zeta_{f,t}$ at time step $t$ are also masked by the attention masks, i.e. $\zeta_{t,k} = \zeta_{f,t} \odot \mathrm{no\_grad}(\mathbf{m}_{t,k})$. The object location $\mathbf{T}_{t,k}$ estimated from the observed points $\mathbf{P}_{t,k}$ is, however, biased toward object surface as the object is only partially observed. We thus use a *where* module that takes $\mathbf{P}_{t,k}$ and $\zeta_{t,k}$ as input and predicts a $\Delta\mathbf{T}_{t,k}$ to correct the bias. To learn a relative translation we first transform $\mathbf{P}_{t,k}$ using the pose parameters $\{\mathbf{T}_{t,k}, \mathbf{I}\}$ as in eq. (6). Concretely, the *where* module is an encoder that consists of several KPConv layers and a KPConv-aggregate layer as the final output layer. The encoded embedding is then fed to a recurrent neural network (RNN) with its hidden states later being decoded into the mean and standard deviation of the posterior distribution $q(\mathbf{z}_{t,k}^{where}|x_{\leq t})$ parameterised as a Gaussian. The $\mathbf{z}_{t,k}^{where}$ is decoded to the $\Delta\mathbf{T}_{t,k}$ through an MLP $f^{where}$ such that:

$$\Delta\mathbf{T}_{t,k} = T_{\max}\tanh(f^{where}(\mathbf{z}_{t,k}^{where})) \tag{7}$$

with $T_{\max} \in \mathbb{R}$ being the maximum delta translation. Given the updated object location $\hat{\mathbf{T}}_{t,k} = \mathbf{T}_{t,k} + \Delta\mathbf{T}_{t,k}$, we encode the shape and the appearance of the object from the observations transformed in the object pose $\{\hat{\mathbf{T}}_{t,k}, \mathbf{R}_{t,k}\}$. Similar to the *where* module, the posterior distribution $q(\mathbf{z}_{t,k}^{what}|\mathbf{z}_{\leq t,k}, x_{\leq t})$ is also parameterised by a KPConv encoder appended by an RNN. If the point number $N_k$ is small, we instead initialise the object pose $\mathbf{T}_{t,k}, \mathbf{R}_{t,k}$ with the object location from the last time step, i.e. $\{\hat{\mathbf{T}}_{t-1,k}, \mathbf{I}\}$ during the propagation.

We use two RNN networks to model the prior distributions $p(\mathbf{z}_{t,k}^{where}|\mathbf{z}_{<t,k}^{where})$ and $p(\mathbf{z}_{t,k}^{what}|\mathbf{z}_{<t,k}^{what})$, whose hidden states are decoded by an MLP to the mean and standard deviation. The prior distribution at $t = 0$ are Gaussian distributions of zero mean for both the $\mathbf{z}^{where}$ and the $\mathbf{z}^{what}$.

## 2.2 Decoder

To explicitly model the 3D geometry of scenes, we represent each object and the background as a Neural Radiance Field (NeRF) (Mildenhall et al., 2020). Each NeRF is a function parameterised by an MLP that maps the world coordinates $\mathbf{p}$ and the viewing direction $\mathbf{d}$ to a color value $\mathbf{c}$ and a density value $\sigma$. One limitation is that a given NeRF can only represent a single scene and cannot condition on the latent encoding. The approach has therefore been extended (Schwarz et al., 2020; Niemeyer & Geiger, 2021) to learn the mapping $f : (\mathbf{p}, \mathbf{d}, \mathbf{z}) \rightarrow (\mathbf{c}, \sigma)$. This introduces a mapping from the latent encoding $\mathbf{z}$ to modifications to the hidden layer outputs of the NeRF MLP. To compose the individual NeRFs into a single scene function, previous approaches (Stelzner et al., 2021; Niemeyer & Geiger, 2021) combine the density values by a summation operation, i.e. $\sigma = \sum_{k=1}^{K} \sigma_k$. This way of composing NeRFs is equal to a superposition of independent Poisson point processes (Stelzner et al., 2021). The scene colour can then be computed as a weighted mean $\mathbf{c} = \frac{1}{\sigma} \sum_{k=1}^{K} \sigma_k \mathbf{c}_k$. We know that a voxel should not be occupied by multiple objects. To account for this property, (Stelzner et al.,

2021) proposes an auxiliary loss to penalize the difference between the sum of the density values and the maximum density values of all components, as simply composing NeRFs with a summation does not suffice. This auxiliary loss, however, requires a warm-up training strategy which introduces additional hyper-parameters for different datasets. We propose to model the scene function as:

$$\sigma = \sigma_{\max} \tanh\left(\sum_{k=1}^{K} \text{softplus}(\sigma_k)\right), \hat{\sigma}_k = \sigma \underset{K}{\text{softmax}}(\sigma_k) \tag{8}$$

In our case, $\sigma_k$ can be interpreted as the logits of the probability, indicating whether this voxel is occupied or not, and $\hat{\sigma}_k$ is the normalised density with a range of $[0, \sigma_{\max}]$. For the colours, we can compute the weighted mean: $\mathbf{c} = \frac{1}{\sigma_{\max}} \sum_1^K \hat{\sigma}_k \mathbf{c}_k$. We upper bound the density by $\sigma_{\max} \in \mathbb{R}$ and model the probabiliy of whether the voxel has been occupied through a tanh. The softplus function ensures that no component can contribute negative occupancy. We use the softmax function to account for the fact that the objects should not overlap, without needing to introduce extra hyper-parameters. To estimate the shape of the objects reconstructed from the NeRFs, we would like a fast approximation approach, as the full evaluation of the volumetric rendering is computationally expensive. Inspired by prior work (Liu et al., 2020), we divide each object bounding box into $\mathbf{S}$ sparse voxels along each dimension. The occupancy at these voxel centres $\mathbf{p}_v$ can then be evaluated by $\sigma_v = \tanh(\text{softplus}(f(\mathbf{p}_v, \mathbf{z}_k)))$. We denote the voxels as occupied if $\sigma_v > \sigma_T$, with the $\sigma_T$ being a threshold. Given the set of the occupied voxels and their centre positions $\mathbf{p}_v$, we can recover the object centre $\mathbf{T}_{t,k}^{\text{shape}}$ and the object orientation $\mathbf{R}_{t,k}^{\text{shape}}$ as discussed in 2.1.2.

## 3 TRAINING

Training a NeRF with known depth requires relatively few evaluations. For example, Stelzner et al. (2021) propose to use only two evaluations of the NeRF for each training iteration, i.e. one evaluation at the surface and one evaluation at points between the camera and the surface. To avoid learning an extremely thin surface, the points sampled at the surface are sampled between the depth range $[\mathbf{d}_{\text{surface}}, \mathbf{d}_{\text{surface}} + \delta]$, with the $\delta$ denoting the surface thickness. The observation loss can be divided into two terms, i.e. a texture loss term:

$$\mathcal{L}_{\text{colour}} = -\log\left(\sum_{k=1}^{K} \mathcal{N}(\mathbf{x}|\mathbf{c}_k, \sigma_{\text{std}}^2) \odot \hat{\sigma}_k^{\text{surface}}/\sigma_{\max}\right) \tag{9}$$

with $\sigma_{\text{std}}$ denoting a fixed standard deviation, and a depth loss term (Stelzner et al., 2021):

$$\mathcal{L}_{\text{depth}} = -\log(\sigma(\mathbf{p}^{\text{surface}})) + \sigma(\mathbf{p}^{\text{air}})/\rho^{\text{air}} \tag{10}$$

where $\rho^{\text{air}}$ is a probability density of the point $\mathbf{p}^{\text{air}}$ being sampled. The latent embedding is regularized using the KL term:

$$\mathcal{L}_{\text{KL}} = \mathbb{KL}(q_\phi(\mathbf{z}_t|\mathbf{z}_{\leq t}, \mathbf{x}_{\leq t})||p_\theta(\mathbf{z}_t|\mathbf{z}_{<t})) \tag{11}$$

An $L2$ loss is used to supervise the *where* module:

$$\mathcal{L}_{\text{where}} = \sum_k (\hat{\mathbf{T}}_{t,k} - \mathbf{T}_{t,k}^{\text{shape}})^2 \tag{12}$$

For the attention mask of the IC-SBP, the learning of the attention masks is supervised via:

$$\mathcal{L}_{\text{att}} = -\left(\log\left(\sum_{k=1}^{K} \mathbf{m}_k \odot (\mathcal{N}(\mathbf{c}_k, \sigma_{\text{std}}) \odot \hat{\sigma}_k^{\text{surface}}/\sigma_{\max})\right) + \log\left(\sum_{k=1}^{K} \mathbf{m}_k \odot \hat{\sigma}_k^{\text{surface}}/\sigma_{\max}\right)\right) \tag{13}$$

Optionally, we found that using a L2 loss for the $\mathcal{L}_{\text{att}}$ can be beneficial for complex 3D scenes, please refer to appendix A.2 for further details. In the end, we penalise the remaining scope $\mathbf{s}_K$ by introducing the remaining scope loss:

$$\mathcal{L}_{\text{scope}} = \sum_{i=1}^{H_1} \sum_{j=1}^{W_1} s_{K,i,j} \tag{14}$$

Taken together, these losses contribute straightforwardly to the overall loss:

$$\mathcal{L} = \mathcal{L}_{\text{colour}} + \mathcal{L}_{\text{depth}} + \mathcal{L}_{\text{KL}} + \mathcal{L}_{\text{where}} + \mathcal{L}_{\text{att}} + \mathcal{L}_{\text{scope}} \tag{15}$$

In training, we use the Adam optimizer with a fixed learning rate of $4e^{-4}$ without any learning rate warm-up or decay strategy.

# 4 EXPERIMENTS

To evaluate OBPOSE's performance on unsupervised object-centric inference and generation of 3D scenes we conduct experiments on the CLEVR-3D dataset (Johnson et al., 2017) and the Multi-ShapeNet dataset used by ObSuRF (Stelzner et al., 2021), and on two YCB object datasets (Calli et al., 2015). One of the YCB object datasets is a moving-object (video) dataset of RGB-D images captured from a fixed front view. The other YCB dataset contains static images captured from three different points of view, where the three views are obtained by rotating the camera by $120°/240°$ around the z-axis of the world frame. All images are shuffled so that ground-truth pairing information is not used for training. In each YCB object dataset, two to four randomly selected objects are spawned on the table. The metrics used for evaluation are described below. Performance on all datasets is compared against the recent baseline of ObSuRF (Stelzner et al., 2021), which, to the best of our knowledge, is the only unsupervised scene inference and generation model that operates on RGB-D images of 3D scenes. To validate our model design decisions, we compare performance using two alternative attention mechanisms in the encoder, one from slot attention (Locatello et al., 2020) and another from an IC-SBP that does not explicitly model object location and orientation (Engelcke et al., 2021). We further compare two ablations of the OBPOSE encoder: one conditioned only on the locations of the objects, and one conditioned on the full 6D poses.

## 4.1 METRICS

In the evaluations, we quantify segmentation quality using the Adjusted Rand Index (ARI) (Rand, 1971; Hubert & Arabie, 1985) and Mean Segmentation Covering (MSC) as metrics. ARI measures the clustering similarity between the predicted segmentation masks and the ground-truth segmentation masks in a permutation-invariant fashion. This is appropriate for unsupervised segmentation approaches where there are no fixed associations between slots and objects. We evaluate segmentation accuracy on foreground objects using the foreground-only ARI and MSC (denoted ARI-FG and MSC-FG, respectively), and on the background using mean Intersection over Union (mIoU). All metrics are normalised between 0 and 1 where a score of 1 indicates perfect segmentation.

## 4.2 UNSUPERVISED 3D OBJECT SEGMENTATION

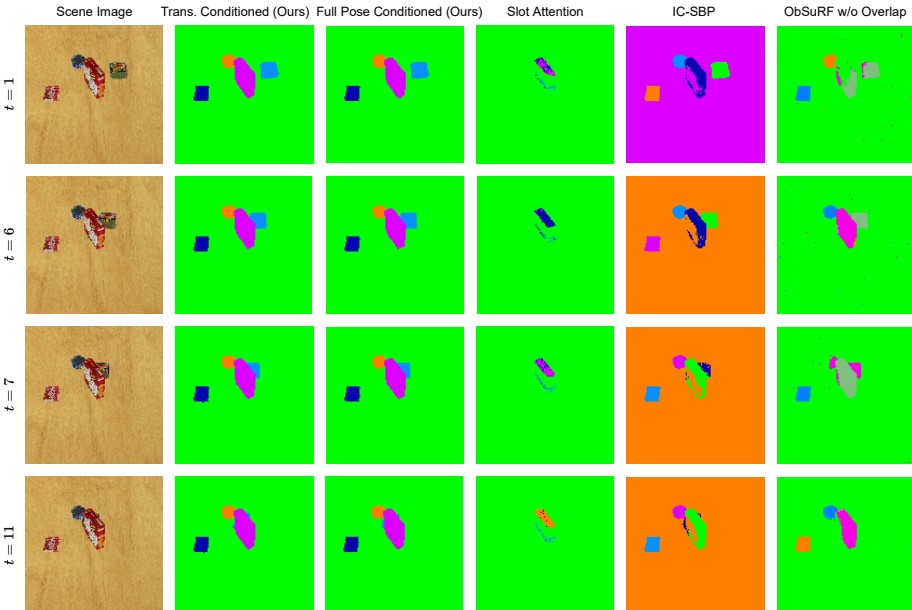

Figure 3: Visualisation of the segmentation results using the YCB moving-object dataset. Our model (OBPOSE) learns better segmentations in the demonstrated video. Each slot is associated with a fixed colour throughout the video. The colours are chosen to be visually distinct from one another.

Table 1: The segmentation results on the CLEVR-3D dataset and MultiShapeNet. The results are rounded to two decimal places.

| | CLEVR-3D | | MultiShapeNet | |
|---|---|---|---|---|
| | ARI-FG↑ | ARI↑ | ARI-FG↑ | ARI↑ |
| OBSURF | 0.96 | **0.95** | 0.81 | 0.64 |
| OBSURF W/O OVERLAP | 0.86 | 0.06 | 0.94 | 0.16 |
| SLOT ATT. | 0.46 | 0.01 | 0.52 | 0.08 |
| OBPOSE (ours) | **0.99** | 0.84 | **0.99** | **0.81** |

Table 2: Mean and standard deviation of the segmentation metrics on the YCB moving-object dataset and the YCB static dataset from three random seeds. The results are rounded to two decimal places. OBPOSE outperforms the baseline and the ablations as it explicitly estimates the locations and the orientations of the objects, disentangling object location and appearance.

| | YCB Moving-Object | | | YCB Static | | |
|---|---|---|---|---|---|---|
| | mIoU-BG↑ | ARI-FG↑ | MSC-FG↑ | mIoU-BG↑ | ARI-FG↑ | MSC-FG↑ |
| OBSURF W/O OVERLAP WITH DEPTH | 0.89 ± 0.09 | 0.18 ± 0.21 | 0.21 ± 0.02 | 0.92 ± 0.04 | 0.31 ± 0.15 | 0.37 ± 0.07 |
| OBSURF W/O OVERLAP | 0.97 ± 0.02 | 0.28 ± 0.36 | 0.28 ± 0.26 | 0.98 ± 0.00 | 0.22 ± 0.12 | 0.36 ± 0.05 |
| SLOT ATT. * | 0.98 ± 0.02 | 0.13 ± 0.05 | 0.19 ± 0.02 | **1.00 ± 0.00** | 0.80 ± 0.01 | 0.84 ± 0.00 |
| IC-SBP | 0.96 ± 0.05 | 0.87 ± 0.02 | 0.90 ± 0.02 | 0.97 ± 0.03 | 0.83 ± 0.09 | 0.80 ± 0.15 |
| OBPOSE NO ROT. (ours) | **1.00 ± 0.00** | **0.96 ± 0.00** | **0.97 ± 0.00** | 0.99 ± 0.00 | **0.89 ± 0.01** | **0.87 ± 0.03** |
| OBPOSE (ours) | **1.00 ± 0.00** | **0.96 ± 0.00** | **0.97 ± 0.00** | 0.98 ± 0.00 | 0.88 ± 0.01 | 0.84 ± 0.03 |

\* The SLOT ATT results are computed with one failed random seed being excluded for the YCB static dataset.

Quantitative results for the CLEVR dataset are shown in Table 1 with qualitative segmentation results in the appendix (Figure 8). We first observe that OBPOSE achieves better foreground segmentation performance (ARI-FG) compared to ObSuRF, which indicates that OBPOSE can perform better scene inference by conditioning on the location and the orientation. The lower full ARI score is caused by the fact that OBPOSE learns to include shadows caused by the existence of the objects in each object slot (see Figure 8), whereas object shadows are labelled as background in the ground-truth masks. Qualitative results on the YCB moving-object dataset are depicted in Figure 3. Quantitative segmentation results are additionally summarised in Table 2. Here we only report ObSuRF results without using the overlap loss as we first observe that the ObSuRF baseline fails to segment the scenes properly using the default setting for 3D data from the open-sourced code. This might be attributed to the weight of a overlap loss used in ObSuRF, to discourage the objects from overlapping. In Stelzner et al. (2021) the same failure mode is reported. In OBPOSE, we instead account for the overlap using a hyper-parameter-free function (*softmax*) in Equation (8). This alleviates the computationally expensive hyper-parameter searching process. Interestingly, using the full 6D pose including the orientation and the location of the objects does not strongly affect the segmentation performance compared to using only the object positions as a way to condition the encoding. This suggests that for object segmentation tasks, the object location itself already provides a strong inductive bias for successful decomposition. Similar results are observed elsewhere in the literature (Kipf et al., 2021), where simple ground-truth position information for objects is used in the first video frame, allowing the model to perform scene segmentation of 2D video data in a weakly supervised fashion. In our approach, we explicitly leverage the 3D reconstruction of the objects whose shape is estimated by the proposed voxelised shape approximation approach. This allows the model to infer the locations and the orientations of objects in a computationally efficient way without using any ground-truth labels. OBPOSE also achieves lower variation on metrics for both datasets, suggesting more stable training compared to the original IC-SBP.

## 4.3 SCENE GENERATION AND SCENE EDITING

Leveraging the disentangled *where* and *what* object-centric representation, OBPOSE can perform more flexible scene generation and scene editing than previous works(Stelzner et al., 2021; Engelcke et al., 2021). We demonstrate this benefit with the CLEVR-3D dataset. In Figure 4(a), we show scenes that are generated from OBPOSE by first sampling from the learned object and the background latent space and then rendering from the composition of object and background NeRFs. The object locations and orientations can be arbitrarily set to user-defined values. Figure 4(b–d) demonstrates that OBPOSE can further be used for flexible scene editing (i.e. adding, removing, or manipulating objects in a generated or inferred scene). The object-level scene manipulation Figure 4(d) as an

important function of OCGMs has raised people's attention in generative models (Niemeyer & Geiger, 2021), OBPOSE thus for the first time implements this in an inference model. Additional view synthesis and segmentation results can be found in the appendix.

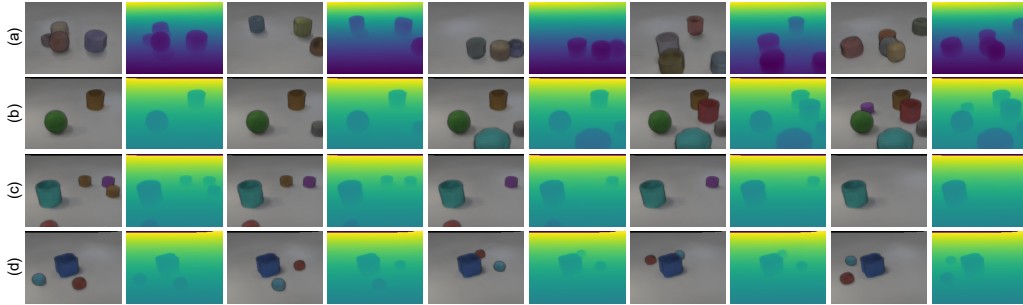

Figure 4: We demonstrate the rendered RGB and depth results of the scenes generated by sampling from the learned latent space (a). We also show the scene editing functions of object addition (b), object removal (c) and object-level scene manipulation (d).

## 5 RELATED WORK

OBPOSE builds upon prior OCGM work on unsupervised segmentation in both 2D and 3D. Most OCGMs for 2D scene segmentation are formulated as variational autoencoders (VAEs) (Kingma & Welling, 2013; Rezende et al., 2014), where different likelihood models serve to explain observations. One set of VAE-OCGMs use bounding boxes, derived from spatial transformer networks (STNs) to represent (*glimpse*) individual objects (Eslami et al., 2016; Xu et al., 2019; Crawford & Pineau, 2019; Lin et al., 2020; Kosiorek et al., 2018; Jiang et al., 2019). Another set represents objects via unsupervised instance segmentation, using pixel-wise mixture models (Burgess et al., 2019; Engelcke et al., 2019; 2020; Greff et al., 2016; 2017; Van Steenkiste et al., 2018; Greff et al., 2019; Veerapaneni et al., 2020; Locatello et al., 2020). This latter set relaxes the spatial-consistency requirements imposed by bounding boxes (Jaderberg et al., 2015), permitting more flexible modelling of objects with complex shapes and textures. However, relaxing spatial consistency has the side-effect that performance can sometimes be biased by features such as the colour of the object (Weis et al., 2020), which has motivated the search for additional inductive biases. A promising candidate is temporal information. To this end, some works (Veerapaneni et al., 2020; Ehrhardt et al., 2020; Kipf et al., 2021) operate on video data and model the correlations between objects explicitly using graph neural networks (GNNs) (Scarselli et al., 2009; Sanchez-Lengeling et al., 2021) or Transformers (Vaswani et al., 2017).

The idea of a reference pose for an object, has precedent in the context of 6D pose estimation, which aims to find the translation and rotation of an object with respect to some frame of reference. In the supervised setting, labels are defined with respect to a given reference frame (Wang et al., 2019; Chen et al., 2020a; Li et al., 2020; Wang et al., 2020; Chen et al., 2020b; Tian et al., 2020). Recently, it has been shown that pose between views of an object, or objects from a common category, can be inferred without labels. Such relative poses have been found for point clouds (Li et al., 2021) and RGB-D images (Goodwin et al., 2022). To the best of our knowledge, we are the first to propose a minimum volume approach for discovering pose without supervision and to include pose information, to reduce its variance in the latent code, as an inductive bias.

## 6 CONCLUSION

We present OBPOSE, an object-centric inference and generation model that learns 3D-structured latent representations. The model extends IC-SBP for video input, and is noteworthy for introducing *poses* as an inductive bias for scene inference. The model's ability to infer object pose is facilitated by several recent innovations, including the use of NeRFs and the fast voxelised shape approximation proposed in this paper. Our experimental results are validated on the CLEVR dataset, the MultiShapeNet dataset, and two synthetic YCB objects datasets. Given its empirical success, outperforming the prior state-of-the-art for static 3D scenes (Stelzner et al., 2021) and establishing a baseline for video, we plan to apply OBPOSE as a vision backbone for robot applications.

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
