# OpenReview forum: "ObPose: Leveraging Pose for Object-Centric Scene Inference and Generation in 3D"
_ICLR.cc/2023/Conference — Submitted to ICLR 2023_

### Official Review · Reviewer_N5Sr · 2022-10-19

**Confidence:** 4
**Correctness:** 3
**Technical Novelty And Significance:** 2
**Empirical Novelty And Significance:** 2
**Recommendation:** 6

**Clarity, Quality, Novelty And Reproducibility:**

This paper is well-written and the overall quality is good. I believe it is reproducible. The novelty is somewhat incremental.

**Strength And Weaknesses:**

Strengths:

1. Compared to the slot-attention-based methods that represent each object as a single latent code, this work explicitly disentangles object location and appearance information, which gives more accurate scene-decomposing results.
2. Ablation study in Table2 justifies its superiority against the slot-attention-based method and IC-SBP-only method.

Weakness:
1. The novelty is limited. (1) The key components such as IC-SBP and canonical pose representation are mostly borrowed from previous methods [7][19]. (2) The idea of representing objects as disentangled location and appearance information is also common, e.g., BlockGAN[R1].
2. The loss function is complicated and the contribution of each component is unclear from the results of the experiments.
3. It would be more convincing if the authors could provide some quantitative comparisons on the unsupervised 6D pose estimation task.

[R1] Nguyen-Phuoc, Thu H., et al. "Blockgan: Learning 3d object-aware scene representations from unlabelled images." Advances in Neural Information Processing Systems 33 (2020): 6767-6778.




**Summary Of The Paper:**

This paper proposes a model OBPose for unsupervised 3D object segmentation from RGB-D images/videos. OBPose first represents each object as disentangled location and appearance information, then re-renders the scene with a NERF decoder. Experimental results prove the effectiveness of the proposed object representation.

**Summary Of The Review:**

I have reviewed this paper in NeurIPS, and the quality of this submission has been improved compared with its previous version. It is nice to see the experiment results and analysis on CLEVR-3D dataset. Therefore, I think it is marginally above the borderline.

---

> ### Author Response · Authors · 2022-11-16
> **Response to reviewer N5Sr**
>
> We thank the reviewer for acknowledging our improvement compared to our last submission.
>
> Reviewer:
>
> “The novelty is limited. (1) The key components such as IC-SBP and canonical pose representation are mostly borrowed from previous methods [7][19]. (2) The idea of representing objects as disentangled location and appearance information is also common, e.g., BlockGAN[R1].”
>
> Authors:
>
> In reply to the reviewer’s concern about novelty, we acknowledge the existence of generative models like BlockGAN. The key idea at the heart of ObPose is that this additional location and orientation information will improve downstream tasks (e.g. segmentation; see the ablation analyses in Table 2). To the best of our knowledge, this is a novel contribution in the context of 3D scene inference and generation.
>
> On a technical level, there are other differences between BlockGAN and ObPose, which represent alternative approaches to important problems in the literature. For example, ObPose detects objects - together with their locations and orientations - from raw observations (pixel colours and depth). Several our key contributions are proposed to enable this for the first time(e.g. shape-based pose estimation, fast shape approximation from NeRFs etc.). However, in a generative model the object pose can be simply sampled from some pre-defined distributions. Unlike BlockGAN, ObPose then uses this information to perform inference (e.g. segmentation). Although ObPose is capable of generating 3D scenes (like BlockGAN), it does so using NeRFs (unlike BlockGAN). Taken together, this makes ObPose both an inference and generation model which has benefits - we can check the models understanding of scenes by looking at generated images while using the object-centric latent space to perform useful tasks like segmentation.
>
> Reviewer:
>
> “The loss function is complicated and the contribution of each component is unclear from the results of the experiments.”
>
> Authors:
>
> We thank the reviewer for flagging this. One way to clarify the losses is to group them. First, there are the observation likelihood loss and the KL-term, which are common to VAEs. Second there is the where loss, which tracks object locations. Finally, there are the attention and scope losses, which are for the IC-SBP. From a practical perspective, the losses are simple to use. That is, simply adding all of the losses works well for all four datasets that we used. This contrasts with the use of auxiliary losses in ObSurF which need to be tuned to each dataset.
>
> Reviewer:
>
> “It would be more convincing if the authors could provide some quantitative comparisons on the unsupervised 6D pose estimation task.”
>
> Authors:
>
> “ObPose is not a pose-estimation model”, nor does it aim at producing pose estimates (quoting from revised introduction). Rather, it uses pose estimates to reduce the variability of observed objects in the model’s latent space (see Figure 11 in the appendix for an illustration). We apologise for any confusion and have tried to add stronger wording early in the paper to clarify this.
>
> We hope with this clarification the reviewer could also agree with us on the point that ObPose is not a model that simply combines existing ideas. Key contributions are proposed to solve problems in inference tasks that do not exist in generative models like BlockGAN. Please let us know if there are other points that still cause the reviewer's concerns, we are ready to try our best to solve these.

---

### Official Review · Reviewer_aWJo · 2022-10-24

**Confidence:** 5
**Correctness:** 2
**Technical Novelty And Significance:** 2
**Empirical Novelty And Significance:** 3
**Recommendation:** 3

**Clarity, Quality, Novelty And Reproducibility:**

**Clarity**: The presentation is quite dense, with the majority of the paper being dedicated to a detailed description of the method. Even so, important aspects, such as the overall probabilistic setup, and the role of voxels, appear along the way without the necessary introduction. Clarity could be improved by focusing on a smaller number of contributions (e.g. cutting the video aspect), abstracting away details into the appendix (such as the specifics of the *where*-module), and adding an overview outlining the general setup. Some of the formulas could use some editing, e.g. using large parenthesis in Eq. 5, and fixing the indices in Eq. 8. Referring to the new synthetic video dataset as "YCB-Video" is misleading, as this term is frequently used for the real world video dataset introduced here: https://rse-lab.cs.washington.edu/projects/posecnn/

**Novelty**: Constructing a 3D model with explicit pose parameters is novel. The paper however seems to miss other recent unsupervised 3D scene understanding methods, namely uORF [1] and OSRT [2] Given the subtle differences between their problem settings, I think it is acceptable to only compare to one baseline, however.

**Reproducibility**: Despite the open questions about the method, the availability of code, and the details in the appendix help reproducibility.

[1] https://arxiv.org/abs/2107.07905

[2] https://arxiv.org/abs/2206.06922

**Strength And Weaknesses:**

Strengths:
 - The paper investigates the interesting question of discovering disentangled 3D representations from RGB-D input.
 - While the possibility of introducing explicit pose parameters was discussed in the ObSuRF paper, to my knowledge, this work is the first to do so. Explicit pose parameters have the potential to simplify object manipulation for downstream tasks.
 - The method shows improved segmentation results on a new synthetic benchmark.

Weaknesses:
 - The provided quantitative results only cover image segmentation results on synthetic benchmarks. Performance on CLEVR3D is similar to ObSuRF. While the model outperforms its baselines on the new YCB images, it remains unclear why new data was generated instead of using the Multishapenet dataset from ObSuRF, given that the two appear to be of similar complexity. In any case, it cannot be claimed that ObPose improves state of the art in 3D segmentation, as the considered datasets are significantly simpler than MSN-hard as used by OSRT [2].
 - The model's most important novel capability, its ability to infer explicit pose parameters, is not directly evaluated. This could be done by comparing to ground truth poses, or computing 3D IoU with respect to ground truth bounding boxes. As a result, it remains unclear if the model's pose parameters function as claimed. Indicating the inferred poses in Figures 3/4 would also help.
 - The softmax formulation in Eq. 8 suppresses the rendering of overlapping objects. However, it does not seem to prevent multiple slots from *representing* the same object with high $\sigma_i$. If these are then rendered separately, e.g. in a scene editing context, the previously suppressed overlaps would emerge, potentially hurting compositionality.
 - The way the paper uses voxels is unclear to me: The term seems to indicate that space is discretized into cubes, but no details are provided on this, and the equations appear to be consistent with standard, continuous NeRF rendering.
 - In general, the paper appears somewhat unfocused, and its presentation could be significantly improved (see below).
 - Contrary to what is stated in this paper, ObSuRF does not use depth information at test time, only as a training signal. ObSuRF's encoder only receives RGB input. Was this changed to allow for a fair comparison?
 - Explicit pose parameters are not strictly necessary for scene editing, as suggested in section 4.3. Adding, deleting and moving objects is also possible with ObSuRF, with the only qualitative difference being that pose parameters allow stating positions in absolute terms, instead of just relative movement. It might be the case that doing this with ObSuRF leads to more artifacts due to imperfect segmentations, however.

**Summary Of The Paper:**

The paper proposes a method for the unsupervised discovery of object-segmented 3D representations with explicit pose parameters, given an RGB-D input image. Building on the ObSuRF model, the proposed ObPose system consists of an encoder inferring pose and shape parameters for a set of objects, and a NeRF decoder which processes these into a set of composable 3D volumes. It is demonstrated that ObPose yields improved segmentation results on new synthetic datasets made up of YCB objects. Additionally, scene editing capabilities are demonstrated.

**Summary Of The Review:**

Overall, while the paper proposes an interesting variant of ObSuRF, there are currently too many issues with the experimental evaluation to recommend acceptance. If the authors can show that the pose parameters work as intended, and that their improvements extend to existing hard benchmarks, I am willing to increase my evaluation.

----

**Post rebuttal update:**
I thank the authors for their response and the updates to the manuscript. I think especially the MSN-easy experiment is an important (and impressive) addition. However, some important issues remain.

**Unfair experimental setup**: The fact that ObPose leverages depths at test time, while ObSuRF doesn't, renders the main experimental results of the paper unfair. The fact that this was swept under the rug in the manuscript is a significant red flag. The added "ObSuRF with depth" baseline cannot fully address this issue:
  * No details were provided on how it was constructed.
  * ObPose's encoder is directly tailored to RGB-D input in many ways, e.g. by using KPConv layers, and by explicitly making use of input point clouds to identify object locations. ObSuRF's encoder is designed for RGB input, so the comparison is still not particularly fair.
  * The more meaningful "ObSuRF with depth" baseline is only provided in Table 2, for the YCB experiment, not in Table 1, for CLEVR3D and MSN.
  * The manuscript still incorrectly states that ObSuRF operates on RGB-D images, and does not make this issue transparent.

I recognize that 3D scene understanding is a developing paradigm in which many problem formulations are possible, and for many of them, there are no appropriate baselines. But these issues need to be treated transparently and put into context. If the authors are looking to argue that their explicit treatment of object location is the key driver of performance for ObPose, an appropriate experiment would be to compare it with an ablation which still leverages RGB-D input, but doesn't contain the pose module. In a setting with RGB-D input, ObSuRF can only provide a lower bound on the performance we may expect.

**Pose Evaluation**: The authors have clarified that the goal of their model is not pose estimation. I agree that there is no need to compare to pose estimation state of the art, or anything like that. However, if the story of the paper is that explicit pose parameters lead to improved segmentation performance, it is still crucial to convince the reader that these pose parameters actually work as advertised. An experiment to that effect is still missing from the main paper.

**Clarity**: No major changes have been made to the structure of the text, as far as I can tell. As a result, my clarity concerns remain.

I believe these issues are too significant to recommend acceptance at this time. I am therefore leaving my score unchanged.

---

> ### Author Response · Authors · 2022-11-16
> **Response to reviewer aWJo**
>
> We thank the reviewer for their helpful comments.
>
> Reviewer:
>
> “Why use YCB dataset instead of MultiShapeNet/”
>
> Authors:
>
> We have added ObPose results on the MultiShapeNet dataset. We find it outperforms the reported ObSuRF results by a large margin (see Table 1).
>
> The YCB dataset (YCB moving-objects - which was previously called YCB-video) is included to see if 3D object-centric generative models can also learn meaningful object-centric representations on single view video data. This is different from other 3D learning models which typically learn from static observations. Nonetheless, the use of a single front-view camera over a table is very common in robot manipulation applications. The success of ObPose on YCB moving-objects is encouraging, as it indicates that 3D OCGMs can learn meaningful object-centric representations (and segmentations) in this practical application setting.
>
> Reviewer:
>
> “The model's most important novel capability, its ability to infer explicit pose parameters, is not directly evaluated. This could be done by comparing to ground truth poses, or computing 3D IoU with respect to ground truth bounding boxes.”
>
> Authors:
>
> As noted in the general comments, ObPose is not a pose-estimation model. It does not aim at producing pose estimates. Rather it uses pose estimates to reduce the variability of observed objects in the model’s latent space (see Figure 11 in the appendix for an illustration). We apologise for any confusion and have added stronger wording in the introduction to clarify this (e.g. “ObPose is not a pose-estimation model”).
>
> Reviewer:
>
> The softmax formulation in Eq. 8 suppresses the rendering of overlapping objects. However, it does not seem to prevent multiple slots from representing the same object with high $\mathbf{\sigma}_i$.
>
> Author:
>
> The tanh term of the $\mathbf{\sigma}$ in eq.8 will first saturate when at least one slot occupies the point and stops generating positive gradients(property of s-shape activation functions). Softmax will then become the only gradient source if there are still multiple slots competing to explain the same point (i.e. positive gradient to the winner and negative gradient to the loser). The positive/negative gradient won’t result in all of the slots having very high $\mathbf{\sigma}_i$.
>
> Empirically, we don’t observe the mentioned problem in the results of the scene generation experiment.
>
> Reviewer:
>
> “The way the paper uses voxels is unclear to me: The term seems to indicate that space is discretized into cubes, but no details are provided on this, and the equations appear to be consistent with standard, continuous NeRF rendering.”
>
> Authors:
>
> You’re right that there are no voxels in the NeRF rendering. However, full shape evaluation with NeRFs is too computationally slow to integrate into the training loop. Voxels are therefore used as part of a fast approximation method. We do this by treating the space within a bounding box as if it were composed of discrete voxels. We then evaluate the occupancy of these cubes to approximate the shape of the object. In practice, even with coarse-grained voxels, we get shape approximations that are both good enough and can be computed on every pass through the training loop.
>
> Reviewer:
>
> “Contrary to what is stated in this paper, ObSuRF does not use depth information at test time, only as a training signal. ObSuRF's encoder only receives RGB input. Was this changed to allow for a fair comparison?”
>
> Authors:
>
> To control for the use of depth information at test time, we also compared our model to a version of ObSuRF that used a depth-conditioned encoder, though we did not find it to significantly affect performance (Table 2).
>
> Reviewer:
>
> “Explicit pose parameters are not strictly necessary for scene editing, as suggested in section 4.3.”
>
> Authors:
>
> We agree with the reviewer, using a relative shift should be able to achieve similar functionality. However, if a slot encodes the object appearance and locations jointly, the generated objects will jump around when the model samples from the learned prior. In ObPose, by contrast, we first sample z_what which only changes the shape and texture of the object before assigning the object with an arbitrary (user-defined) pose. We believe being able to disentangle appearance and location is a desirable feature to have.
>
> Reviewer:
>
> "YCB-Video" is misleading, as this term is frequently used for the real world video dataset(washington).
>
>  Authors:
>
> We have renamed the dataset in the paper so that it is now the “YCB moving-object dataset”, to avoid confusion.
>
> Reviewer:
>
> “Some of the formulas could use some editing, e.g. using large parenthesis in Eq. 5, and fixing the indices in Eq. 8”
>
>  Authors:
>
> Thank you. We have updated these equations.
>
> We thank the reviewer for giving such high quality reviews that help us improve our paper. If there are any further concerns, please let us know, we will try our best to address them.

---

### Official Review · Reviewer_uLft · 2022-10-24

**Confidence:** 4
**Correctness:** 3
**Technical Novelty And Significance:** 3
**Empirical Novelty And Significance:** 2
**Recommendation:** 5

**Clarity, Quality, Novelty And Reproducibility:**

The paper is well written and method is clearly described. Related work also covers the most relevant works in the literature to my knowledge. Perhaps the authors should be more precise about what type of 6D pose their method is claimed to estimate, as there are a number of different perspectives in the literature.
The combination of IC-SBP, estimation of where and what latents, variational inference, and object-centric NeRF decoding is certainly novel (although none of these methods are new). Their main technical novelty is the specific bound-box volume method for pose estimation and the way they use it to efficiently voxelize a NeRF object and use it to refine the pose. This can be a useful contribution to the literature but as per the weaknesses above, further experiments would help clarify the significance of this novelty wrt the model's performance.
Finally, I believe the method is presented in sufficient detail to be reproducible, and the datasets used are well known.

**Strength And Weaknesses:**

**Strengths**
- The use of strong inductive biases allows this self-supervised model to obtain representations suitable for domains in which detailed scene understanding is required. These biases are: object-centric segmentations, inference of disentangled 3D pose and shape, and the ability to represent objects in explicit 3D voxels.
- The authors propose a novel method for shape representation via voxelization using the "minimum volume principle", an efficient approach that allows end-to-end training.
- Results highlight the model's ability to obtain precise 3D (and 2D) segmentations from just one or a few images, and to consistently outperform closely related work in this task on CLEVR-3D and YCB datasets.

**Weaknesses**
1. While the minimal bounding-box volume selection is an interesting heuristic for pose estimation, it's unclear from the paper if the method proposed learns consistent canonical poses across instances of the same category. Related work [1, 2] in self-supervised pose estimation methods ultimately evaluate their method against some form of GT category-level poses. This work would be strengthened if similar evidence is provided (in the form of additional evaluations against GT poses) or if at least extensive examples of estimated poses are shown. Note that if the above estimated poses are not consistent, the fast shape evaluation may also fail to produce consistent shapes across instances of the same category.
2. Experimental results suggest that using the inferred rotations do not seem to help, which weakens the significance of one of the paper's novelties (i.e. the benefits of the where->what pipeline). That said, shape estimation is used to refine the object location prediction via $L_{where}$. Thus, for the whole approach to be an interesting contribution, the authors should show that location refinement (delta T) is important to the overall model performance.
3. Finally, I am concerned regarding the fairness of the comparison with ObSurf. In the most challenging YCB datasets, the authors only report ObSurf w/o overlap loss. In ObSurf's paper it is shown to be an important loss. In my eyes, for this comparison to be fair, the authors should perform a reasonable hyper-parameter search and not just use ObSurf's "default settings for 3D". Usually optimal settings may vary for each dataset and indeed ObPose's settings also change between CLEVR-3D and YCB as per Table 3 in the appendix.

**Questions / suggestions**

- What happens if $N_k$ is small in the first time step?
- Requiring depth in general is a stronger requirement than many methods in the literature, including ObSurf which only uses depths for training signal. Due to ObPose's point-cloud approach, it cannot be trained without depths, but for the sake of fairness the authors can compare with ObSurf that additionally conditions the encoder with depths.
- What are failure modes of the method? It would be enlightening to see more challenging examples i.e. clutter and occlusion from YCB.
- How do $L_{att}$ and $L_{scope}$ (which don’t seem essential) affect performance?
- Also, since $L_{att}$ is different for some datasets, this should be stated in the main text.
- Is the IC-SBP baseline exactly ObPose but without the use of pose-based transformation of features?
- The use of tanh activation to compute the global density means densities can be negative. Are these the densities the ones used for volumetric rendering? If so breaks the non-negativity assumption of volumetric rendering.

[1] Li et al. Leveraging SE(3) Equivariance for Self-supervised Category-Level Object Pose Estimation from Point Clouds. NeurIPS 2021

[2] Zero-Shot Category-Level Object Pose Estimation. ECCV 2022

**Summary Of The Paper:**

This work proposes an object-centric 3D inference method that is trained in a self-supervised way using a variational auto-encoder framework and neural radience fields. While a large number of methods in the literature address object-centric self-supervised segmentation, very few also learn to infer 3D pose and shape while doing so. The authors propose using the existing video-specific instance colouring stick-breaking process to encode the input imagines into segmented attention masks, and propose a novel mechanism to refine and estimate each detected object's pose. Given disentangled latents pose (where) and content (what), novel images can be rendered using a NeRF MLP for any viewpoint. Once trained ObPose obtains near-perfect segmentations on a number of datasets and is shown to outpeform ObSurf as well as a number of other baselines.

**Summary Of The Review:**

The direction of this work is important for robotics and this method can be a useful way to learn detailed representations from images without requiring large datasets of segmented objects. The combination of techniques is novel and the authors propose a new heuristic to estimate and refine the pose of the segmented objects. Further clarification on the method's ability to detect pose is important, as well as additional experiments wrt. ablations of some losses and a fairer comparison with ObSurf. I believe these points are necessary to establish the significance of this work. As it is now, I am leaning towards rejecting it.

---
**Post rebuttal update:**

I thank the authors for addressing my questions and concerns, and for updating the manuscript with improvements in a wide range of aspects. Most importantly, ensuring a fairer comparison with ObSurf as well as the addition of another dataset. I believe the paper is in a better state now, but I still think it's insufficient to raise my score due to the fundamental limitations below.

As reviewer aWJo also shares, two of my main concerns not been fully resolved after the rebuttal.

First, it is still unclear to me what the benefits are of the pose refinment novelty via the minimum volume principle. As acknowledged by the authors, conditioning the encoding on rotation doesn't result in clear benefits (while conditioning on location does); and neither have the authors provided evidence that the location refinement (delta T) is important to the overall model performance. It is also still unclear if the poses reflect consistently the real poses of the objects which would justify the hypothesis the authors refer to regarding lower variance of appearance representation. In other words, if poses estimated are arbitrary for different instances of similar classes (e.g. chairs of different shape but same appearance) why should we expect the appearance representations of them to be invariant?

Second, while the comparison with ObSurf is now fairer, it is still difficult to assess this work in the wider 3D scene understanding literature.
This method requires RGB-D whereas most unsupervised segmentation methods do not. Methods such as ObSurf and Object Scene Representation Transformers (Sajjadi et al. 2022) work without depth and also are able to produce great reconstructions (especially the latter). ObPose's renders are fairly poor (Fig. 7) which casts doubt about how well it can work (i.e. that relies on reconstructions) on the increasingly richer and more realistic datasets such as MultiShapenet-Hard, e.g. scenes with heavy clutter and complex textures. While this may be improved with better latent representations of z_what and z_where and a stronger NeRF decode, all in all it would involve substantial changes to the method and paper.

---

> ### Author Response · Authors · 2022-11-15
> **Response to reviewer uLft--part 1**
>
> We thank the reviewer for the constructive criticism which we have tried to incorporate into the revised draft.
>
> Reviewer:
>
> “it's unclear from the paper if the method proposed learns consistent canonical poses across instances of the same category. Related work [1, 2] in self-supervised pose estimation methods ultimately evaluate their method against some form of GT category-level poses. This work would be strengthened if similar evidence is provided (in the form of additional evaluations against GT poses) or if at least extensive examples of estimated poses are shown.”
>
> Authors:
>
> As mentioned in the general comments, there appears to be some confusion about the role of pose information in our model, and we apologise if the name “ObPose” contributed to that. ObPose is not a pose estimation model but rather a scene segmentation, generation, and editing model. As such it is distinct from papers that output pose estimates, such as those cited by the reviewer [1, 2].
>
> Our hypothesis in developing the model is that the inclusion of pose information should reduce the variance of object-centric latent representations(see Figure 11 in the appendix). Intuitively, we would like a less variant latent representation for an object in different poses.
>
> We find that pose information does improve performance (see “ObPose” in Table 2), compared to ablations of the model without pose information (“IC-SBP”), but the benefits came mainly from conditioning on object locations rather than orientations (cf. “ObPose” and “ObPose no rot”).
>
> We agree, exploring pose information used by the model could be a valuable way to check our assumption about the quality of pose information being used. We have therefore added information about pose used by the model to Figure 11 in the appendix.
>
> Reviewer:
>
> “Experimental results suggest that using the inferred rotations do not seem to help, which weakens the significance of one of the paper's novelties (i.e. the benefits of the where->what pipeline). That said, shape estimation is used to refine the object location prediction via Lwhere. Thus, for the whole approach to be an interesting contribution, the authors should show that location refinement (delta T) is important to the overall model performance.”
>
> Authors:
>
> As the reviewer observes, “shape estimation is used to refine the object location prediction”. Another way to think about it is that the ablation with orientation (“ObPose no rot” in Table 2) argues for the object location itself being a strong inductive bias (see discussion  in section 4.2). We note complementary results elsewhere in the literature [e.g. Kipf et al., 2021] where an object’s ground-truth position (but not orientation) is used in the first video frame.
>
> Reviewer:
>
> “I am concerned regarding the fairness of the comparison with ObSurf. In the most challenging YCB datasets, the authors only report ObSurf w/o overlap loss. In ObSurf's paper it is shown to be an important loss. In my eyes, for this comparison to be fair, the authors should perform a reasonable hyper-parameter search and not just use ObSurf's "default settings for 3D".”
>
> Authors:
>
> We share this concern and appreciate the suggestion. To represent ObSuRF as fairly as we can, we have therefore been running additional hyperparameter searches on the YCB dataset. In a further effort, we have added a new experiment to the paper using the MultiShapeNet dataset, which ObSuRF was evaluated again in the original paper. We find that ObPose outperforms ObSuRF’s reported numbers on the MultiShapeNet dataset by a large margin (see Table 1).

---

> ### Author Response · Authors · 2022-11-15
> **Response to reivewer uLft--part 2**
>
> Reviewer:
>
> What happens if Nk is small in the first time step?
>
> Authors:
>
> This indicates a severe occlusion. If it happens, we set the orientation of the object to be the same as the world frame. ObPose is not a pose-estimation model. The goal is not to predict object pose but to leverage the pose for better inference. If no meaningful orientation can be inferred from the shape, we simply condition on the object location.
>
> Reviewer:
>
> “Requiring depth in general is a stronger requirement than many methods in the literature, including ObSurf which only uses depths for training signal. Due to ObPose's point-cloud approach, it cannot be trained without depths, but for the sake of fairness the authors can compare with ObSurf that additionally conditions the encoder with depths.”
>
> Authors:
>
> We thank the reviewer for this suggestion. Because of it, we have added an ablation study to the paper where we modify the ObSuRF encoder so that it is also conditioned on depth (see Table 2). But we find that this does not help much on performance.
>
> Reviewer:
>
> What are failure modes of the method?
>
> Authors:
>
> An important failure mode of ObPose is that our model can not predict meaningful object orientations from the shape when the object is severely occluded. In this case, we thus do not condition the encoding on the object orientation but only on the object location. Also, we found that the wooden box in the YCB static dataset is segmented as part of the background. We hypothesise that this is due to it having a similar texture to the wooden table (background). However, the wooden box is segmented correctly in the YCB video dataset, which would appear to suggest that the temporal structure of video can improve object segmentations.
>
> Reviewer:
>
> How do L_att and L_scope (which don’t seem essential) affect performance?
>
> Authors:
>
> These are essential. L_att is used to supervise the IC-SBP and the attention mechanism learns to cluster point-wise embeddings into objects. L_scope penalises the remaining scope acquired after the last step of the IC-SBP, so that all observations can be explained by previous slots. Performance decreases dramatically without either of these losses.
>
> Reviewer:
>
> “Also, since L_att is different for some datasets, this should be stated in the main text.”
>
> Authors:
>
> This was stated in the appendix but has now been moved to the main text.”
>
> Reviewer:
>
> “Is the IC-SBP baseline exactly ObPose but without the use of pose-based transformation of features?”
>
> Authors:
>
> Yes.
>
> Reviewer:
>
> The use of tanh activation to compute the global density means densities can be negative. Are these the densities the ones used for volumetric rendering? If so breaks the non-negativity assumption of volumetric rendering.
>
> Authors:
>
> We use a softplus function before the tanh. Consequently, the input of tanh is positive and the output density will also be all positive; see Eq. (8).

---

### Official Review · Reviewer_opyj · 2022-10-25

**Confidence:** 5
**Correctness:** 4
**Technical Novelty And Significance:** 3
**Empirical Novelty And Significance:** 2
**Recommendation:** 5

**Clarity, Quality, Novelty And Reproducibility:**

I think the paper is overall nicely written. Some section can be improved but
in general it is easy to follow. To the best of my knoweledge the proposed
model is novel and I think reproducing the presented results won't be an issue.

**Strength And Weaknesses:**

## Strengths:
------------

1. The idea of disentangling the concept of "where to look" from "what to look"
is very interesting and is applicable to various tasks beyond oibject segmentation.

2. I also liked the author's solution for finding th object's pose using the
minimum volume principle.

3. I liked the additional results on scene editing in Section 4.3. However,
this section seems a bit unconnected with the rest of the text. So it might be
good to revise the text to better prepare the reader for this experiment.

4. I appreciate that for the CLEVR experiment, the authors also visualize the
corresponding depth maps, since their model can also predict them.

5. Overall the proposed idea is simple, novel and achieves very good results.

## Weaknesses:
-------------
1. The experimental evaluation is a bit weak. First of all, I think that the
datasets that the authors use for evaluating their model are rather simple,
namely the YCB dataset consists only of two scenes and the CLEVR has very
simply object arrangements. However, since also ObSuRF considers such a simple
experimental setup I am eager to overlook this. Nevertheless the experiments
could be improved. For example,  is there a reason why there is no comparison
with slot attention on the CLEVR dataset (see Table 1.)? Furthermore, can the
authors provide some additional qualitative comparison with ObSuRF and Slot
Attention also on the CLEVR dataset.

2. Looking at the original results from the ObSuRF paper, their segmentations
look quite plausible. Hence, I find it a bit weird that for the YCB dataset it
fails completely. Therefore, I think it is important to show a similar
comparison as in Figure 3 also for the CLEVR dataset.

3. A less important weakness of this paper in comparison to slot attention is
that it assumes that the number of objects in the scene is known. While I
believe that this constraint does not impose such a strong limitation it is
still a limitation that is useful to be addressed.

4. Since the authors only consider very simple datasets in their evaluation, I
am wondering how generalizable this model is to more complex scenes.


**Summary Of The Paper:**

This paper addresses the task of unsupervised object segmentation from RGB-D
video data.  In particular, the authors introduce an object-centric
representation that is used to decompose each RGB-D image into a set of NeRFs
per scene object. Note the camera poses from which the RGB-D video data were
captured are considered to be known. The RGB-D input, is first converted into a
pointcloud which is then passed into a KPConv backbone, that extracts per-point
features. These features are subsequently used to compute the background
embedding and the soft attention masks per object, which are produced by
clustering the per-point features. Given these masks, they then predict the
pose of each object. The location of each object is simply the center of mass
of the points within each cluster, and the rotation is computed following the
minimum volume principle. Finally, conditioned on the object poses, they
extract features that are passed to a per-object NeRF. The proposed model is
evaluated on the task of unsupervised scene segmentation on the CLEVR dataset
and on the YCB dataset using both RGB-D videos and multi-view static scenes.
The authors consider two baselines: The ObSurf and slot attention and
demonstrate that their method outperforms prior works in terms of various
metrics such as the mean Intersection over Union (mIOU), the Adjusted Rand
Index (ARI) and the Mean Segmentation Covering (MSC).


**Summary Of The Review:**

Overall, I like the idea of this paper but I am a bit skeptical regarding the experimental evaluation. I think it is essential to provide the additional results I mentioned above on the CLEVR dataset in order to ensure a stronger submission. Below are some more comments/questions and suggestions:

1. Symbol $\zeta_f$ does not appear in Figure 2. I recommend to add in Figure 2
to avoid confusion. From the description in the last sentences of Section 2.1
it is not 100% clear how $\zeta_f$ and $\zeta_c$ are computed from $\zeta$.

2. Can the authors provide some more details regarding the computation of
$\mathbf{b}z^{\textbf{bg}}$ from $\zeta$? I think that providing additinal details on
this would significantly improve clarity.

3. If I understand correctly the Instance Colouring
Stick-Breaking Process, described in Section 2.1.1 is simply a clustering
mechanism. I believe that it would significantly improve the paper's clarity if the authors
added one/two sentences in the beginning of this section pointing out this
fact. Starting with all the specific details make understanding quite hard. In
particular, regarding section 2.1.1, I believe that the authors should revisit
the text and try to provide some intuition regarding the things they describe.
For example, why is the stochastic ordering of the mask important? I think that
giving intuitive examples always facilitates understanding.

4. From Section 2.1 it is not clear that the feature embedding $\zeta_f$ refers
to point features at a particular time step t (see Section 2.1.2). This is
quite confusion and should be clarified in Section 2.1

5. The reconstruction results in Figure 7 in the supplementary seem to have
some artifacts. I am wondering to the authors use also importance sampling in
their NeRF, or do they only rely on a coarse NeRF model?

6. How sensitive is the IC-SBP module to initialization?

---

> ### Author Response · Authors · 2022-11-15
> **Response to Reviewer opyj**
>
> We thank the reviewer for the thorough feedback and constructive criticism. We updated the paper to address several shortcomings.
>
> Reviewer:
> “The experimental evaluation is a bit weak. First of all, I think that the datasets that the authors use for evaluating their model are rather simple, namely the YCB dataset consists only of two scenes and the CLEVR has very simply object arrangements. However, since also ObSuRF considers such a simple experimental setup I am eager to overlook this. Nevertheless the experiments could be improved. For example, is there a reason why there is no comparison with slot attention on the CLEVR dataset (see Table 1.)? Furthermore, can the authors provide some additional qualitative comparison with ObSuRF and Slot Attention also on the CLEVR dataset.”
>
> Authors:
> To address this concern, we have added a new experiment which uses the hard MultiShapeNet dataset used by ObSuRF. The results outperform ObSuRF on this dataset by a large margin(0.81 vs 0.64 for the ARI score), as shown, for instance, in the requested comparison figure (Figure 10). We find that the slot-attention ablation model does not perform well on the CLEVR and MultiShapeNet datasets(see Table 1.). This suggests that without the key contribution (where->what) of ObPose, and thus only naively combining existing modules (slot attention, NeRF) from previous works, does not lead to satisfying performance.
>
> Reviewer:
> “A less important weakness of this paper in comparison to slot attention is that it assumes that the number of objects in the scene is known. While I believe that this constraint does not impose such a strong limitation it is still a limitation that is useful to be addressed.”
>
> Authors:
> We are grateful to the reviewer for flagging a lack of clarity in our text, and thus for giving us an opportunity to address it. ObPose does not require the number of objects to be specified as an input to the model. Rather, one specifies a number of available slots for objects, which do not need to be filled. This is the same as in Slot Attention.
>
> Reviewer:
> “Below are some more comments/questions and suggestions…”
>
> Authors:
> We thank the reviewer for proposing helpful improvements to the paper.
>
> For Q1, we have followed the reviewer’s suggestion and modified Figure 2 accordingly. The two heads are two point-wise MLPs (explanation added to section 2.1).
>
> For Q2, we compute the ζ using a U-Net-like encoder-decoder structure. In order to re-use the encoding, we employ an aggregation layer. This takes the last layer output of the encoder as input and aggregates the result into z_bg.
>
> For Q3, we have added the sentences suggested by the reviewer at the beginning of section 2.1.1. The reason for not discussing the intuition of IC-SBP is that this clustering mechanism has already been done in previous work where it was proposed [GENESIS-v2], and is thus not our key contribution. Within the alotted space, we focused instead on discussing the IC-SBP and how it contributes to the logic of the system as a whole.
>
> For Q4, we have added the sentences suggested by the reviewer to clarify this.
>
> For implementation details mentioned in Q1and Q2, we believe that the clearest and most efficient answer we can give is to direct the reader to our open-sourced codebase (please refer to the mentioned git repository in the appendix file).
>
> For Q5, we use importance sampling.
>
> For Q6, we refer the reviewer to the GENESIS-v2 paper, which contains details about IC-SBP. IC-SBP is not our key contribution, however, as we mention in response to Q3.
>
> We have also made a number of changes to the text to improve clarify where indicated by the reviewer.
>
> Finally, if there are any other points to address, please let us know and we will do our best to address them. We appreciate the overall positive tone of the review and would be grateful if the reviewer would consider adjusting his or her score.

---

### Author Response · Authors · 2022-11-15
**General comments to all reviewers**

We thank the reviewers for all the useful feedback. We also apologise for an apparent lack of clarity around the role played by “pose” in the paper. A number of reviews seem to view ObPose as a “pose-estimation model” – which it is not. We have therefore added the following text to the introduction: “ObPose is not a pose-estimation model. Rather, ObPose infers pose information from an object's shape to reduce appearance variance and to simplify the learning of the model’s what component in 3D (see Figure 11 in appendix) – ultimately for use in downstream tasks like segmentation.”

A second point that appeared across multiple reviews concerned novelty. A number of papers are similar to our work in some ways, but different in others. For example, RELATE[1] is generative and does scene editing, but does not do inference/segmentation. SAVI++[2] does segmentation in video and uses depth, but is not generative and does not do scene editing. (SAVI[3] is like SAVI++ but without depth.) OSRT[4] and uORF[5] do segmentation, view synthesis (extrapolating from given views), and scene editing, but are not generative and do not generate novel scenes or operate in video. ObPose does all of these things - inference, segmentation, scene generation, scene editing, and video - within a single model. There is therefore a richness to the shared object-centric representations in its latent space. Most similar in spirit to our model is ObSuRF, which does 3D scene segmentation and, although not posed as such, could be used for scene generation. We compare against ObSuRF explicitly.

The more major changes to the paper include: (1) the addition of a new dataset (MultiShapNet) and evaluations on it; (2) ablations of the ObSuRF encoder with depth input; (3) slot attention ablation on the CLEVR-3D and MultiShapNet datasets; and (4) additional hyper-parameter search on the overlap loss.

[1] RELATE: Physically Plausible Multi-Object Scene Synthesis Using Structured Latent Spaces

[2] SAVi++: Towards End-to-End Object-Centric Learning from Real-World Videos

[3] Conditional Object-Centric Learning from Video

[4] Object Scene Representation Transformer

[5] Unsupervised Discovery of Object Radiance Fields

---

### Decision · Program_Chairs · 2023-01-20

**Decision:**

Reject

**Justification For Why Not Higher Score:**

There are multiple concerns including the fairness of the comparison (e.g., whether depth is needed as input), the quality of the results, the technical novelty, and the presentation.

**Justification For Why Not Lower Score:**

N/A

**Metareview: Summary, Strengths And Weaknesses:**

The paper aims to learn a factorized scene representation, separately encoding object location (where) and appearance (what). The reviewers agree that this is a hard and important problem to solve and appreciate the initial results on the simple dataset. However, there are a few important issues that remain after the rebuttal and the discussion. These issues include the fairness of the comparison (e.g., whether depth is needed as input), the quality of the results, the technical novelty, and the presentation. Eventually, reviewers and the AC agree that this paper is promising but below the bar.  The authors are encouraged to revise the paper accordingly for the next venue.